# Proteomic Determinants of Variation in Cholesterol Efflux: Observations from the Dallas Heart Study

**DOI:** 10.3390/ijms242115526

**Published:** 2023-10-24

**Authors:** Anamika Gangwar, Sneha S. Deodhar, Suzanne Saldanha, Olle Melander, Fahim Abbasi, Ryan W. Pearce, Timothy S. Collier, Michael J. McPhaul, Jeremy D. Furtado, Frank M. Sacks, Nathaniel J. Merrill, Jason E. McDermott, John T. Melchior, Anand Rohatgi

**Affiliations:** 1Department of Internal Medicine, Division of Cardiology, University of Texas Southwestern Medical Center, Dallas, TX 75390, USA; anamika.gangwar@utsouthwestern.edu (A.G.); sneha.deodhar@utsouthwestern.edu (S.S.D.); suzanne.saldanha@utsouthwestern.edu (S.S.); 2Department of Clinical Sciences, Lund University, 221 00 Malmö, Sweden; olle.melander@med.lu.se; 3Department of Medicine, Division of Cardiovascular Medicine, Stanford University School of Medicine, Stanford, CA 94305, USA; fahim@stanford.edu; 4Quest Diagnostics Cardiometabolic Center of Excellence, Cleveland HeartLab, Cleveland, OH 44103, USA; ryan.w.pearce@questdiagnostics.com (R.W.P.); timothy.s.collier@questdiagnostics.com (T.S.C.); 5Quest Diagnostics Nichols Institute, San Juan Capistrano, CA 92675, USA; michael.j.mcphaul@questdiagnostics.com; 6Department of Nutrition, Harvard T.H. Chan School of Public Health, Boston, MA 02115, USA; jfurtado@hsph.harvard.edu (J.D.F.); fsacks@hsph.harvard.edu (F.M.S.); 7Biogen Inc., Cambridge, MA 02115, USA; 8Biological Sciences Division, Pacific Northwest National Laboratory, Richland, WA 99354, USA; nathaniel.merrill@pnnl.gov (N.J.M.); jason.mcdermott@pnnl.gov (J.E.M.); john.melchior@pnnl.gov (J.T.M.); 9Center for Lipid and Arteriosclerosis Science, Department of Pathology and Laboratory Medicine, University of Cincinnati, Cincinnati, OH 45237, USA; 10Department of Neurology, Oregon Health and Science University, Portland, OR 97239, USA

**Keywords:** atherosclerotic cardiovascular disease, cholesterol efflux capacity, high-density lipoproteins (HDLs), apolipoprotein, proteomics

## Abstract

High-density lipoproteins (HDLs) are promising targets for predicting and treating atherosclerotic cardiovascular disease (ASCVD), as they mediate removal of excess cholesterol from lipid-laden macrophages that accumulate in the vasculature. This functional property of HDLs, termed cholesterol efflux capacity (CEC), is inversely associated with ASCVD. HDLs are compositionally diverse, associating with >250 different proteins, but their relative contribution to CEC remains poorly understood. Our goal was to identify and define key HDL-associated proteins that modulate CEC in humans. The proteomic signature of plasma HDL was quantified in 36 individuals in the multi-ethnic population-based Dallas Heart Study (DHS) cohort that exhibited persistent extremely high (>=90th%) or extremely low CEC (<=10th%) over 15 years. Levels of apolipoprotein (Apo)A-I associated ApoC-II, ApoC-III, and ApoA-IV were differentially correlated with CEC in high (r = 0.49, 0.41, and −0.21 respectively) and low (r = −0.46, −0.41, and 0.66 respectively) CEC groups (p for heterogeneity (pHet) = 0.03, 0.04, and 0.003 respectively). Further, we observed that levels of ApoA-I with ApoC-III, complement C3 (CO3), ApoE, and plasminogen (PLMG) were inversely associated with CEC in individuals within the low CEC group (r = −0.11 to −0.25 for subspecies with these proteins vs. r = 0.58 to 0.65 for subspecies lacking these proteins; *p* < 0.05 for heterogeneity). These findings suggest that enrichment of specific proteins on HDLs and, thus, different subspecies of HDLs, differentially modulate the removal of cholesterol from the vasculature.

## 1. Introduction

Atherosclerotic cardiovascular disease (ASCVD) is the biggest cause, globally, of mortality and the economic healthcare burden [1]. In essence, ASCVD results from an imbalance of lipid deposition and the removal of cholesterol from the vasculature. This removal is mediated by high-density lipoproteins (HDLs) in a process called reverse cholesterol transport (RCT), in which lipids are transported from the vasculature to the liver for excretion [2,3,4]. It was long thought that plasma measures of HDL cholesterol (HDL-C) reflected RCT efficiency and atheroprotection. This notion was called into question when studies emerged showing that lipid-modifying therapies did not correlate with change in HDL-C levels [5] and that genetically determined low HDL-C levels are not causally associated with increased risk of ASCVD [6,7]. These reports, followed by the well-publicized failure of pharmaceutical interventions to improve ASCVD outcomes, despite successfully raising HDL-C in individuals [8,9,10,11,12], revealed the major flaws in reliance of HDL-C as a reflection of HDL functionality. This resulted in a major shift in the HDL field to focus on other compositional features of HDLs that impart functionality.

Remarkably, HDLs have been shown to have pleiotropic functional impacts on different metabolic pathways within the vasculature [13,14], but their role in RCT is still thought to be their primary cardioprotective function. The first critical step of RCT is the efflux of cholesterol from lipid-laden macrophage foam cells in the vasculature to HDLs via membrane transporters (ATP binding cassette transporter A1 (ABCA1), ATP binding cassette transporter G1 (ABCG1), and scavenger receptor-B1 (SR-B1)) [3]. Multiple human studies have shown this functional property of HDLs—termed cholesterol efflux capacity (CEC)—is inversely associated with atherosclerosis and incident CVD [15,16,17,18,19,20,21,22]. Importantly, these associations are independent of traditional cardiometabolic risk factors and lipid measures and provide strong evidence that CEC is a more robust and clinically relevant marker for cardiometabolic phenotype than traditional measures of HDL-C. To date, the molecular basis of variation in CEC remains unknown.

The pleiotropic functionality of HDLs is mirrored by their compositional complexity, with studies now demonstrating that these particles can associate with over 250 different proteins. These proteins can generally be segregated into two types: (i) dynamic scaffold proteins, such as apolipoprotein (apo)A-I, that act as detergents to solubilize the lipid for transport and (ii) accessory proteins that can dock with the scaffolds or with exposed lipid to impart specific functions. Scaffold proteins usually comprise most of the protein on the particle—for instance, apoA-I, the primary scaffold protein on HDLs, is reported to make up 70–80% of the total protein. Conversely, other accessory proteins are thought to make up only a small percentage of the protein, but they impart discrete unique functions to different subspecies of HDLs. The HDL-speciation profile of an individual can vary, based on multiple physiological, genetic, dietary, and environmental cues, and changes in these profiles and proteomic signatures have been previously associated with changes in CEC [23,24].

Conventional proteomics methods report total abundance of a given protein but do not provide comprehensive information about the concentration of a given subspecies of HDLs. Further, HDL-speciation procedures can be time consuming and difficult to scale [25,26], limiting their application in large clinical studies. Recently, an enrichment method using an affinity-purifiable lipid-free apoA-I coupled with targeted proteomics was developed as a high-throughput method for analyzing proteins that associate with apoA-I [27]. The HDL apolipoproteomic score (pCAD) calculated using this method was associated with the presence of coronary artery disease (CAD), independent of circulating apoA-1 and other conventional cardiovascular risk factors [28]. On the other hand, in a multi-cohort study, Furtado et al. recently developed a novel enzyme-linked immunosorbent assay (ELISA) that can rapidly and precisely quantify native plasma apoA-I containing lipoprotein (AI-Lp) subspecies [29], based on the functional accessory proteins associated with the particles, and showed that these protein-based subspecies are associated with higher relative risk of coronary heart disease (CHD) [30]. Our goal was to apply these high-throughput proteomic assays to our well-defined clinical population to identify potential components on the HDLs that contribute to CEC.

We previously established that extremely low (<10th%) and extremely high (>90th%) CEC is a persistent trait in individuals over a period of 15 years in the Dallas Heart Study (DHS) cohort [31]. In that study, we reported that total HDL protein and apoA-I concentrations were similar in individuals that contained HDLs with either extremely high or extremely low CEC [31], indicating that another component of the particle is likely mediating differences in CEC. Given the reported diversity of the HDL proteome, we hypothesized these accessory proteins that define different HDL subspecies [32] are responsible for the differences in the CEC of HDLs observed in our study. To investigate this, we applied complementary HDL-speciation and proteomic characterization approaches to molecularly profile apoA-I-associated proteins and minor AI-Lp subspecies in individuals with persistent and extremely low or extremely high CEC. Using a novel computational approach, we integrated the proteomic data across different methods to identify key regulatory proteins associated with the CEC of HDLs.

## 2. Results

### 2.1. Study Population Demographics and Clinical Characteristics

The demographics and clinical characteristics of the participants with persistent-extremely high or extremely low CEC (Appendix A) were described previously [31] and are reported briefly in Table 1 and Table 2. The median age of the participants was 60 years, with similar percentages of female and Black participants across the two groups (Table 1). Traditional cardiovascular risk factors were not significantly different between the extremely high and extremely low CEC groups (Table 2), except that participants in the low CEC group had significantly lower triglyceride levels than participants in the high CEC group (median low 97 mg/dL versus median high 120 mg/dL), *p* = 0.04; Table 2). The low and high CEC groups were not significantly different in HDL-C levels (median low 55 mg/dL versus median high 48 mg/dL) or apoA-I levels (median low 155 mg/dL versus median high 154 mg/dL) (*p* > 0.05 for both, Table 2).

### 2.2. ApoA-I-Associated Proteins Are Heterogeneously Correlated with Plasma Cholesterol Efflux Capacity (CEC) in Extremely High versus Extremely Low CEC Groups

We investigated apoA-I-associated proteins by “dipping” lipid-free apoA-I into plasma samples. The apoA-I contained a histidine tag that allowed for the isolation of the protein and bound material, using affinity enrichment. These proteins were subsequently quantified, using targeted proteomics. The abundances of apoA-I-associated proteins were not different between persistent extremely high and extremely low CEC groups (Appendix A). However, when we performed a correlation analysis, we observed an association of certain apoA-I-associated proteins with plasma CEC but not with the fractionated CEC area under curve (CEC-AUC) (Appendix A) within the extremely high or extremely low CEC groups (Appendix A). ApoC-II was positively correlated with plasma CEC in the high CEC group (r = 0.49, *p* = 0.04), while vitronectin and apoA-IV were positively correlated with plasma CEC in the low CEC group (r = 0.58, 0.66; *p* = 0.020, and 0.01, respectively).

We observed opposite correlations between certain apoA-I-associated proteins and plasma CEC when comparing the persistent extremely high CEC and extremely low CEC groups. p heterogeneity tested the significance of differences in the protein-CEC correlations between the high CEC and low CEC groups. ApoC-II and apoC-III were positively correlated (r = 0.49 and 0.41, respectively) with CEC in the high group but negatively correlated (r = −0.46 and −0.41, respectively) with CEC in the low group (p for heterogeneity (pHet) = 0.03 and 0.04, respectively). ApoA-IV was positively correlated (r = 0.66) with CEC in the low group and negatively correlated (r = −0.21) with CEC in the high group (pHet = 0.003) (Figure 1).

### 2.3. ApoA-I Containing Lipoprotein (AI-Lp) Subspecies with versus AI-Lp Subspecies without Certain Proteins Display Heterogeneous Correlations with Fractionated CEC in the Extremely Low CEC Group but Not in the Extremely High CEC Group

Next, we turned to a novel ELISA that measured the abundance of AI-Lp subspecies, defined by presence or absence of specific functional proteins of interest; i.e., these were a collection of native plasma lipoproteins that all contained apoA-I but were differentiated by the presence or absence of an accessory protein X. As was the case in the lipid-free A-I dipping assay, we observed no differences in the abundance of AI-Lp subspecies between persistent extremely high and extremely low CEC groups (Appendix A). The main finding is the heterogeneity in correlation between CEC and AI-Lp subspecies, with or without a protein (Figure 2).

AI-Lp subspecies were correlated with fractionated CEC but not with plasma CEC (Appendix A). We speculate that fractionation of plasma may result in the enrichment of specific subspecies relative to other components in the plasma and modulate its relationship with efflux. Since we observed a correlation of AI-Lp subspecies only with CEC-AUC, we pursued further heterogeneity testing only on AI-Lp subspecies-fractionated CEC correlations. In the low CEC group, AI-Lp subspecies lacking apoC-III, complement C3 (CO3), apoE, and plasminogen (PLG) were each associated positively with the fractionated CEC area under curve (CEC-AUC) (r = 0.55 to 0.65), while A1-Lp subspecies with these proteins were not correlated with CEC-AUC (r = −0.08 to −0.25; *p* < 0.05 for heterogeneity) (Figure 2(Aa,Ba,Bb,Bc,Bd), Appendix A).

In the high CEC group, no significant heterogeneity was found in correlations between CEC and AI-Lp subspecies that contained or lacked the same proteins (Figure 2(Ab)).

### 2.4. Relationships between apoA-I-Associated Proteins and apoA-I Containing Lipoprotein (AI-Lp) Subspecies Display Heterogeneity between the Extremely High and Extremely Low CEC Groups

Next, we performed a deep biostatistical analysis across all datasets to extract more nuanced relationships between the proteome and CEC. To do so, we applied a protein–protein pairwise correlation analysis using the python package pandas performed on the apoA-I-associated proteins and AI-Lp subspecies in the high and low CEC groups. We observed distinct protein–protein correlation clusters between the high and low CEC groups (Figure 3). In the high CEC group, we observed one major cluster, marked as cluster H1, showing correlation between multiple apo proteins, SAA4 from targeted proteomics for apoA-I-associated proteins, and AI-Lp subspecies (apoA-I with the protein and apoA-I without the protein). Notably, AI-Lp subspecies (e.g., apoA-I with A2M and apoA-I without A2M) correlated with whole plasma apoA-I (WPAI from ELISA for AI-Lp subspecies) and apoA-I (from targeted proteomics for apoA-I-associated proteins) in the high CEC group (Figure 3a).

In the low CEC group, we observed three major clusters, marked as clusters L1, L2, and L3, respectively. Cluster L1 showed correlation between multiple “apoA-I with protein” subspecies along with SAA1/2. Cluster L2 contained multiple “apoA-I without protein” subspecies and WPAI (whole plasma ApoA-I). In Cluster L3, multiple apoA-I-associated proteins (apoL-I, ANGT, apoA-II, A1AT, PON1, apoD, apoM, and VTNC) were correlated (Figure 3b). For protein–protein correlation analysis, the major difference observed between the high and low CEC groups was that in the high CEC group there was clustering of all the AI-Lp subspecies (apoA-I with and without proteins) and WPAI/apoA-I in the same cluster (cluster H1), while in the low CEC group there was separate clustering of “apoA-I with proteins” and “apoA-I without proteins” (cluster L1 and cluster L2, respectively).

## 3. Discussion

CEC, as a clinically relevant trait for ASCVD, independent of HDL-C and other risk factors, has been validated by most but not all epidemiological studies [15,16,17,18,19,20,21,22,33]. Our own investigations established that CEC was linked to incident ASCVD events and that extremely low (<10th%) and extremely high (>90th%) CEC was a persistent trait over 15 years in the same cohort (DHS) [31], thus lending evidence that CEC is a robust and clinically relevant cardiometabolic phenotype. A promising anti-atherosclerotic strategy, supported by mechanistic animal studies and epidemiologic human studies, is to increase CEC [17,34,35,36,37]. However, we have yet to discover what are the molecular drivers or causal factors determining CEC variation, which may serve as targets for manipulation for improved cardiometabolic outcomes.

Using advanced proteomic phenotyping of apoA-I-associated proteins and AI-Lp subspecies, we found heterogeneous protein–CEC correlations between extremely high and extremely low CEC groups. Abundances of total apoA-I, 27 apoA-I-associated proteins, and AI-Lp subspecies were similar in the high and low CEC groups. However, despite similar abundances, correlations with CEC were heterogeneous in the high and low CEC groups for certain apoA-I-associated proteins, including apoC-II, apoC-III, and apoA-IV. In addition, correlations between CEC and AI-Lp subspecies were almost completely blunted in the presence of certain proteins, including apoC-III, CO3, apoE, and PLG in the low CEC group.

Collectively, our data showed heterogeneity in advanced proteomic correlates of CEC, despite a lack of differences in abundances. These observations suggested that simply measuring protein abundances in whole plasma is not sufficient to completely understand the molecular determinants of CEC variations and that deeper analysis may be required. The heterogeneity for correlations between CEC and apoA-I-associated proteins and AI-Lp subspecies indicated that factors other than abundances like protein modifications, proteoforms, and subspecies lipid composition might be playing an important role in CE variation.

Although the link between HDL proteome and HDL function is evident [38], the effect specific HDL-associated proteins and HDL subspecies exert on CEC is not completely known. Multiple studies, including our own, have reported the total protein and apoA-I abundance to be similar across CEC phenotypes. Thus, other HDL-associated proteins and HDL subspecies could be functionally relevant for CEC and explain the link between CEC and ASCVD. Our current study focuses on this premise and, building upon prior literature correlating HDL proteins to CEC, we now show heterogeneity in the association between CEC and apoA-I-associated proteins (apoC-II, apoC-III, and apoA-IV) and AI-Lp subspecies (apoC-III, CoC3, apoE, and PLG) that is responsible, in part, for variation in CEC.

Findings from previous studies support the impact of apoA-I-associated proteins on CEC. In vitro, the HDL proteome (with ~250 proteins) has been shown to modulate CEC via cholesterol transporters, lipoprotein lipids, and cholesterol acceptor proteins, with apoA-II, apoE, and plasminogen playing important roles [39,40]. ApoA-IV has been reported to promote CEC in cultured cells [41,42,43]. ApoC-III on the contrary has been reported in multiple studies to hamper CEC [44,45,46]. ApoC-II has unexplored aspects to its effects on CEC. He et. al. reported that apoC-II in small HDLs is negatively correlated with ABCA1-mediated CE in type 2 diabetes [47]. Liu et al. reported that apoC-II-KO zebrafish suffered severe hypertriglyceridemia and were rescued by administering human apoC-II mimetic peptide/plasma from wild-type (WT) zebrafish [48]. A genome-wide association study by Low-Kam et al. reported that variants in locus *apoE/C1/C2/C4* were major determinants of CEC, independent of HDL-C [49]. Our study extends these findings to a human cohort with extreme and persistent high and low CEC and reveals novel heterogeneity by CEC groups. ApoA-I-associated apoC-II, apoC-III, and apoA-IV were differentially correlated with CEC in both the high and low CEC groups. Given the extreme and persistent CEC trait in this study cohort, these findings provide biological relevance for these apoA-I-associated proteins in CEC variation.

Many minor HDL subspecies defined by the presence or absence of a specific protein play an active role in HDL functions like cholesterol transport and are associated with CVD [29,30,50,51,52]. These AI-Lp subspecies comprise about 10% or less of the total apoA-I concentration of HDL. In the low CEC group, the correlation between CEC and AI-Lp subspecies was almost completely blunted when specific proteins were present in these subspecies (apoC-III, CoC3, apoE, or PLG). This supports the notion that low-abundance subspecies, defined by protein content, are functionally relevant potential molecular drivers of CEC. The enrichment of apoA-I containing subspecies with apoC-III impairs HDL atheroprotective functions, including CEC, and is associated with increased risk of CVD [44,53]. ApoE content of apoA-I-containing subspecies strongly predicts CVD risk [54,55,56,57] and reduces the HDL’s CEC via the ABCA1 pathway [58]. These observations indicate that the presence of apoC-III and apoE impairs CEC and may be responsible for impaired HDL cardioprotective function. ApoA-I containing subspecies that contain CO3 or PLG were associated with a higher risk of CHD than the complementary subspecies that lack the protein [30]. Plasmin reduces HDL-induced cholesterol efflux from macrophages [59]. Another study reported that PLG promoted cholesterol efflux by the ABCA1 pathway [60]. CO3 modulates the immune response but its role in CEC is not clear. However, various other complementary factors have been associated negatively with CEC [61]. Our study showed that the presence of PLG and CO3 on apoA-I-containing subspecies leads to a loss of correlation with CEC, suggesting a loss of atheroprotection. Such divergent findings necessitate future studies to understand the role of HDL-associated PLG and CO3 in driving variations in HDL functions.

Bioinformatics, using protein–protein correlation analysis, revealed separate clustering of most of the apoA-I-associated proteins and AI-Lp subspecies, indicating that the two methodologies used in this study—affinity enrichment using apoA-I and ELISA for Lp-AI subspecies—yield drastically different information about HDL particles. This also suggests that these minor AI-Lp subspecies are distinct particles, and studying only HDL-associated protein abundances is not sufficient to understand the complete biology and role of apoA-I-containing particles in CEC variation.

We report heterogeneity in the CEC-protein correlations between the high and low CEC groups for apoA-I-associated proteins and AI-Lp subspecies, despite similar abundances of these proteins and subspecies. How these proteins modulate CEC is unclear, but our findings indicate the possibility of different arrangements of these proteins on AI-Lp subspecies across the HDLs’ size range that may affect the particles’ affinity for cholesterol transporters on the plasma membrane. Further investigations focused on the key proteins identified in this study can provide valuable insight into the molecular mechanisms of CEC variation. In this study, we focused only on the extremes of CEC (high and low). Future studies are needed to compare the extreme CEC distribution to the middle zone of CEC distribution in order to understand the complete mechanism of cholesterol efflux 

Cholesterol efflux is relevant for various cardiometabolic diseases, such as ASCVD, diabetes, and chronic kidney disease. Therefore, determining the molecular factors that regulate cholesterol efflux variation may have broad implications for risk assessment and for identifying novel targets for prevention and treatment. Future studies are warranted, including the functional characterization of the identified HDL subspecies, to determine causality and the degree of effect on CEC; to identify other protein features (e.g., proteoforms) and their relevance to CEC; and to further identify the deep phenotyping of the proteome of minor HDL subspecies that most potently modulate CEC.

## 4. Materials and Methods

### 4.1. Study Population

An extreme and persistent high and low CEC cohort was previously established and described [31]. This cohort comprised a subset of participants from the Dallas Heart Study (DHS)—a multiethnic, population-based cohort [62]. Briefly, CEC was measured in 2924 participants enrolled in the DHS at the baseline visit (DHS1 2000–2002) [22]. A random subset of participants below the 10th or above the 90th percentile of CEC distribution were prospectively recruited (N = 57; 2017–2020), of which 36 participants were determined to have persistently high or low CEC (Figure 4). Participants with malignancy, HIV, chronic kidney disease (CKD), and pregnancy were excluded.

The study protocol was approved by the Institutional review Board of the University of Texas Southwestern Medical Center and was conducted in compliance with institutional guidelines.

### 4.2. Blood Collection and Storage

Blood samples were collected in standard blood collection tubes, maintained for ≤3 h at 4 °C and, then, centrifuged for 15 min at 3000 rpm at 4 °C to isolate plasma (BD 366643) and serum (367988). Plasma and serum were collected, aliquoted in cryotubes, and stored at −80 °C.

### 4.3. Affinity Enrichment and Targeted Proteomics Analysis of ApoA-I-Associated Lipoproteins

All apoA-I-associated lipoproteins measurements were performed on blinded samples at the Quest Diagnostics Cleveland HeartLab (Cleveland, OH, USA), using a recently described and validated method [27]. Metal chelate affinity chromatography was used to isolate ApoA-I-associated lipoproteins from de-identified human serum, using a Freedom Evo automated liquid handler (Tecan, Männedorf, Switzerland) for end-to-end automation. Briefly, recombinant ^15^N-His_6_ApoA-I in 1X PBS, pH 7.4, was added to human serum, incubated, diluted, and, then, purified using PhyTips (Phynexus, San Jose, CA, USA), packed with Ni-NTA HisBind Superflow stationary phase. The sample was bound to the Phytip columns, followed by washing with 20 mM imidazole, 20 mM sodium phosphate, and 150 mM sodium chloride, pH 8.0. Subsequently, resin-bound His6-ApoA-I and associated proteins were eluted with 300 mM imidazole and 50 mM Tris-HCl, pH 9.0, 25% methanol. The eluted samples were then denatured using heat treatment, followed by digestion with endoproteinase LysC (Wako Chemicals USA, Richmond, VA, USA) and ^13^C_6_, ^15^N_2_-lysine-labeled internal standard peptides (Vivitide, Gardner, MA, USA and Life Technologies, Carlsbad, CA, USA) addition. An Agilent 6495A triple-quadrupole mass spectrometer operating in dynamic MRM mode was used for detecting peptides from digested samples, allowing for the targeted detection of peptide targets from ApoA-I, ApoA-II, ApoA-IV, ApoC-I, ApoC-II, ApoC-III, ApoC-IV, ApoD, ApoE, ApoL-I, ApoM, Alpha-1-antitrypsin (A1AT), angiotensinogen (ANGT), cholesteryl ester transfer Protein (CETP), clustering (CLUS), complement-CIII (CO3), haptoglobin (Hp), kallistatin (KAIN), lecithin cholesterol acyltransferase (LCAT), lipoprotein-associated phospholipase-A2 (LpPLA2), phospholipid transfer protein (PLTP), serum paraoxonase/arylesterase-1 (PON1), serum paraoxonase/arylesterase-3 (PON3), retinol-binding protein-4 (RET4), serum amyloid alfa-1/2 (SAA1/2), serum amyloid alfa-4 (SAA4), transthyretin (TTHY), and vitronectin (VTN). Two transitions per peptide and up to 2 peptides per protein were monitored. MassHunter v10.1 (Agilent, Santa Clara, CA, USA) was used for quantitative analysis to obtain peptide signal intensities via integrating the chromatographic peak for the quantifier transition.

### 4.4. Measurement of ApoA-I-Containing Subspecies with and without Selected Protein (AI-Lp Subspecies)

A novel modified sandwich ELISA, developed by Sacks et al., was used to measure the concentration of the AI-Lp subspecies defined by the protein’s presence (the concentration of apoA-I in HDLs with protein) or absence (the concentration of apoA-I in HDLs without protein). We selected 7 proteins based on their role in the range of HDL functions: alpha-2-macroglobulin (A2M), apoC-I, apoC-III, apoE, complement C3 (CO3), haptoglobin (Hp), and plasminogen (PLG). This protocol was described in detail previously [29]. Briefly, seven 96-well microplates were prepared, each one coated with a different antibody corresponding to one of the 7 apoA-I containing subspecies-defining proteins. De-identified human plasma samples diluted in PBS were loaded in duplicate in each microplate and incubated overnight at 4 °C. The unbound depleted fraction of the apoA-I-containing subspecies that contained the defining protein was removed, followed by 3 washes with PBS. Tween-containing diluent was loaded into the wells to dissociate the bound lipoprotein complexes. Anti-apoA-I antibody-coated 96-well plates were also prepared, one for each type of apoA-I-containing subspecies. The unbound fraction containing apoA-I subspecies lacking the defined protein (apoA-I without protein) and the dissociated bound fraction containing the apoA-I subspecies with the defined protein (apoA-I with protein) from the previous steps was transferred to these anti-apoA-I antibody-coated plates and the quantity of the bound apoA-I, with or without protein, was determined by sequential incubations with biotinylated anti-apoA-I, streptavidin, and o-phenylenediamine substrate. The absorbance was measured at 450 nm.

### 4.5. Statistical Analysis

Statistical analyses were performed using GraphPad Prism (GraphPad Software, Inc., La Jolla, CA, USA, version 8.0) and SAS (version 9.4M7). *p* values for comparison of demographics and risk factors between the high and low groups were calculated using the χ^2^ test for categorical variables and the Kruskal–Wallis test for continuous variables. For comparison of the apoA-I associated proteins and AI-Lp subspecies abundances of the extremely low and extremely high CEC groups, a 2-tailed Student *t* test was used. Correlation analysis was used to assess the association between ApoA-I-associated proteins/A1-Lp subspecies and CEC measures (CEC of apoB depleted plasma (plasma CEC) and the fractionated CEC area under curve (CEC-AUC). Heterogeneity testing was used to analyze for heterogeneity of protein–CEC correlations. For ApoA-I associated proteins, from 27 proteins we selected the proteins with the maximum difference (∆) in the correlation coefficient (r) between the high and low CEC groups, with a cutoff of −0.2 to 0.2 for ∆. Correlations of these selected proteins with efflux measurements (plasma CEC and CEC-AUC) were tested for the heterogeneity of the high and low CEC groups. For AI-Lp subspecies, the heterogeneity of the correlations between apoA-I with proteins and efflux measurements (plasma CEC, CEC-AUC) from correlations between apoA1 without proteins and efflux measurements (plasma CEC, CEC-AUC) in the high and low CEC groups were analyzed. A two-sided *p* value ≤ 0.05 was considered significant.

To assess the protein profiles of patients divided into the low and high CEC groups, we used the Pearson correlation coefficient from the pandas library (Version 1.4.4), which measures linear dependency between protein values within their respective groups. For each protein pair, the correlation analysis was conducted independently for the low and high CEC groups. In addition, any instances of missing values were excluded from the correlation analysis. Proteins that had associated correlation coefficients exceeding the threshold of 0.7 (*p* < 0.01) were considered significant relationships.

## Figures and Tables

**Figure 1 ijms-24-15526-f001:**
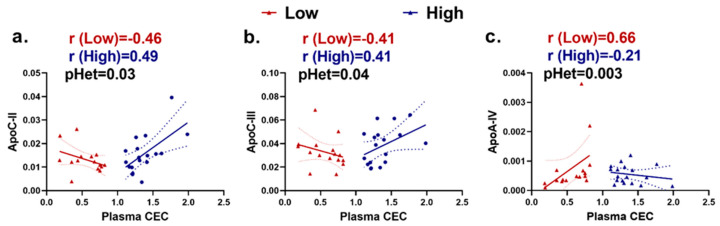
ApoA-I-associated proteins showed heterogeneous correlation with plasma cholesterol efflux capacity (CEC) in individuals with extremely low and extremely high CEC. Scatter plots show heterogeneity in (**a**) ApoC-II, (**b**) ApoC-III, and (**c**) ApoA-IV correlation with plasma CEC. The low CEC group is represented in red and the high CEC group is represented in blue. ApoC-II: apolipoprotein C-II, ApoC-III: apolipoprotein C-III, ApoA-IV: apolipoprotein A-IV.

**Figure 2 ijms-24-15526-f002:**
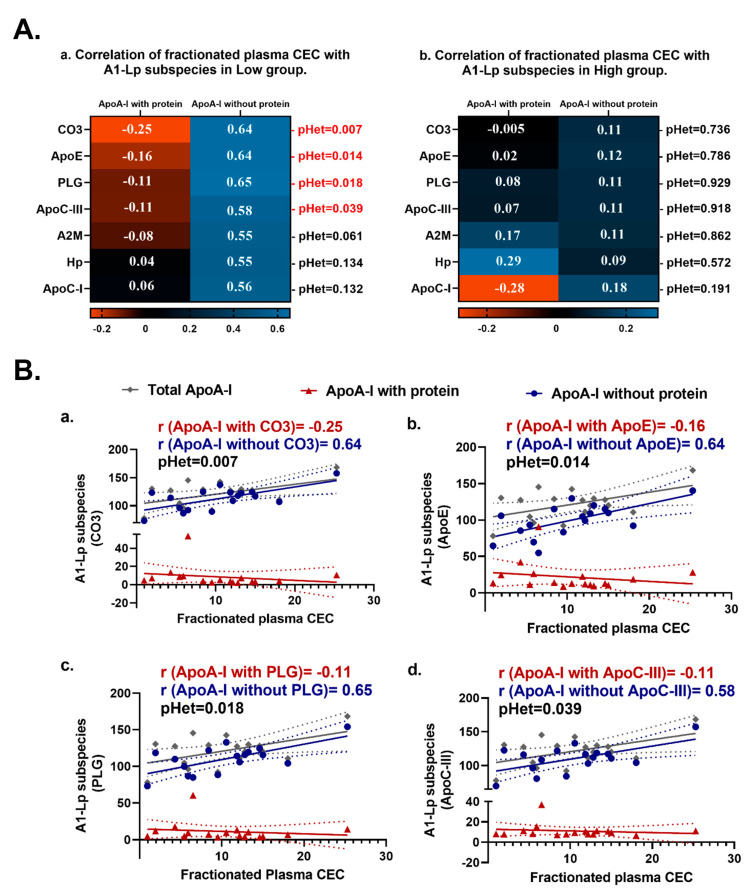
Association between apoA-I containing lipoprotein (AI-Lp) subspecies (ApoA-I with and without certain proteins) and the CEC area under the curve (AUC) in extremely low and extremely high CEC. (**A**) Heatmap showing heterogeneity between AI-Lp subspecies (ApoA-I with protein and ApoA-I without protein). (**Aa**) Correlation of AI-Lp subspecies with fractionated CEC in the low CEC group; (**Ab**) correlation of AI-Lp subspecies with fractionated CEC in the high CEC group. Red represents a negative correlation with CEC and blue represents a positive correlation with CEC. CEC-AUC was derived from CEC measured in size-exclusion chromatography (SEC) fractionated plasma fractions using superpose 6 column (10/300 GL; GE Healthcare) on an ÄKTA pure protein purification system (GE Healthcare). (**B**) Scatter plots for selected proteins showing blunting of correlation between AI-Lp subspecies and fractionated CEC with the presence of certain proteins in the low CEC group. (**Ba**) AI-Lp subspecies with or without C3, (**Bb**) AI-Lp subspecies with or without ApoE, (**Bc**) AI-Lp subspecies with or without PLG, (**Bd**) AI-Lp subspecies with or without ApoC-III. AI-Lp subspecies with a certain protein (ApoA-I with protein) are represented in red, AI-Lp subspecies without a certain protein (ApoA-I without protein) are represented in blue, and total ApoA-I is represented in grey. CO3: complement C3, ApoE: apolipoprotein E, PLG: plasminogen, ApoC-III: apolipoprotein C-III, A2M: alpha-2-macroglobulin, Hp: haptoglobin, ApoC-I: apolipoprotein C-I, ApoA-I: apolipoproteinA-I.

**Figure 3 ijms-24-15526-f003:**
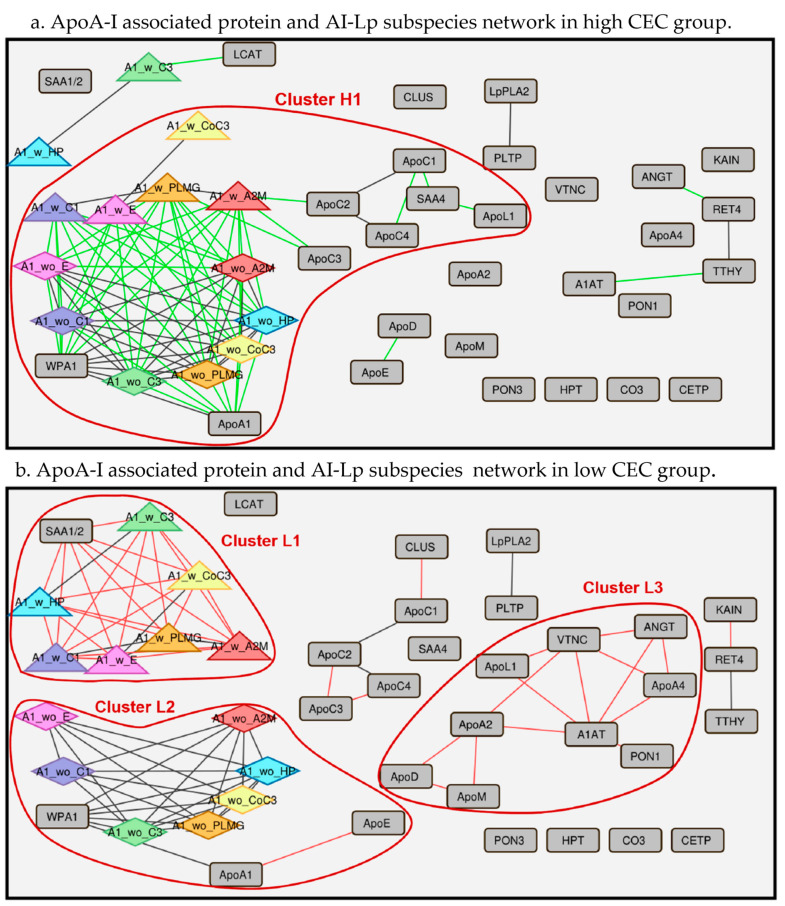
Protein–protein correlation network containing apoA-I-associated protein and apoA-I containing lipoprotein (AI-Lp) subspecies, showing heterogeneity between extremely high and extremely low CEC individuals. (**a**) Protein–protein correlation network between apoA-I-associated proteins and AI-Lp subspecies in the high CEC group. The green line represents the unique associations present in the high CEC group. The black line represents the common associations observed in both the high and low CEC groups. (**b**) Protein–protein correlation network between apoA-I-associated proteins and AI-Lp subspecies in the low CEC group. The red line represents the unique associations present in the low CEC group. The black lines represent the common associations observed in both the high and low CEC groups. A1 with protein represents AI-Lp subspecies defined by the presence of a protein (e.g., A1_w_C3, A1_w_A2M). A1 without protein represents AI-Lp subspecies defined by the absence of a protein (e.g., A1_wo_C3 and A1_wo_A2M).

**Figure 4 ijms-24-15526-f004:**
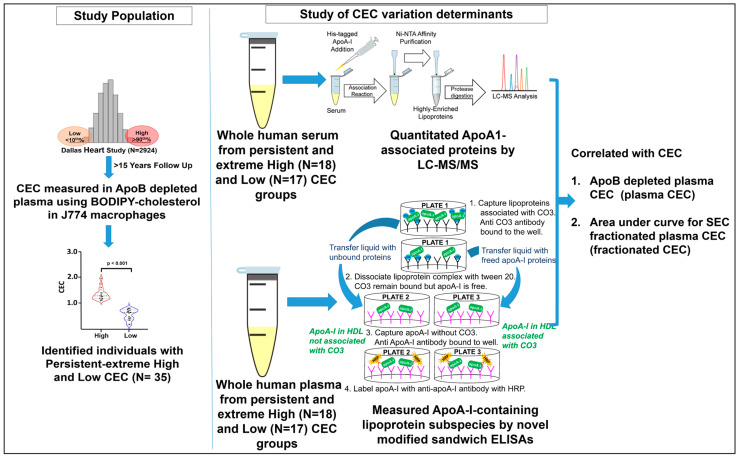
Study design. CEC: cholesterol efflux capacity, ApoB: apolipoprotein B, ApoA-I: apolipoprotein A-I. Whole human serum/plasma samples were used from persistent-extremely low and high CEC cohorts (i.e., the included participants from the Dallas Heart Study) and subjected to targeted proteomics for the identification and quantification of ApoA-I-associated proteins and to novel modified ELISA for the quantification of seven A1-Lp subspecies. The data obtained were further analyzed for their associations with CEC.

**Table 1 ijms-24-15526-t001:** Demographics of participants with persistent extremely high and extremely low cholesterol efflux capacity (CEC). Continuous variables are reported as medians (interquartile range) and categorical variables are reported as percentages (N); *p* values were calculated using the Kruskal–Wallis test for continuous variables and the χ^2^ test for categorical variables.

Variable	Persistent High Efflux (N = 19)	Persistent Low Efflux (N = 17)	*p*-Value
Age, years	56 (51–67)	64 (58–71)	0.12
Female sex (%)	68% (13)	71% (12)	0.89
Hispanic ethnicity (%)	21% (4)	6% (1)	0.24
Black race (%)	63% (12)	59% (10)	0.79

**Table 2 ijms-24-15526-t002:** Clinical (self-reported) and biochemical variables of participants with persistent extremely high and extremely low cholesterol efflux capacity (CEC). Continuous variables are reported as medians (interquartile range) and categorical variables are reported as percentage (N); *p* values were calculated using the Kruskal–Wallis test for continuous variables and the χ^2^ test for categorical variables. BMI: body mass index; HDL-C: high-density lipoprotein cholesterol; LDL-C: low-density lipoprotein cholesterol.

Variable	Persistent High Efflux (N = 19)	Persistent Low Efflux (N = 17)	*p*-Value
Hypercholesterolemia (%)	50% (9)	53% (9)	0.86
Diabetes (%)	33% (6)	24% (4)	0.52
Heart disease (%)	11% (2)	0% (0)	0.15
Hypertension (%)	44% (8)	71% (12)	0.12
Menopausal (% of women)	84% (11)	92% (11)	0.59
Current smoker (%)	21% (4)	6% (1)	0.19
History of alcohol intake (%)	68% (13)	71% (12)	0.89
Blood pressure medication (%)	47% (9)	69% (11)	0.20
Glucose-lowering medication (%)	26% (5)	25% (4)	0.93
Lipid medication (%)	42% (8)	37% (6)	0.78
BMI	30.4 (26.3–35.4)	29.8 (26.0–34.6)	0.96
Total cholesterol, mg/dL	178 (159–247)	154 (146–197)	0.27
LDL-C, mg/dL	102 (82–163)	88 (81–126)	0.21
Non-HDL-C, mg/dL	120 (103–198)	107 (95–146)	0.09
Triglycerides, mg/dL	120 (97–226)	97 (75–117)	0.04
HDL-C, mg/dL	48 (41–58)	55 (49–61)	0.27
Apolipoprotein A-I, mg/dL	154 (148–182)	155 (149–171)	0.92
Glucose, mg/dL	103 (89–125)	95 (90–104)	0.59
Hemoglobin A1C, %	5.7% (5.5–6.8)	5.8% (5.5–5.9)	0.61
Creatinine, mg/dL	0.88 (0.70–0.98)	0.81 (0.58–0.92)	0.33
Total protein, g/dL	7.3 (7.0–7.5)	7.3 (6.9–7.3)	0.38
Albumin, g/dL	4.4 (4.2–4.5)	4.5 (4.1–4.6)	0.34
Hemoglobin, g/dL	13.6 (12.9–14.9)	13.7 (13.0–14.3)	0.81

## Data Availability

The data presented in this study are available on request from the corresponding author.

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
