# Peer review of "Proteomic Determinants of Variation in Cholesterol Efflux: Observations from the Dallas Heart Study"

_ijms, 2023, doi:10.3390/ijms242115526_

Round 1

Reviewer 1 Report

Proteomic Determinants of Variation in Cholesterol Efflux: Observations from the Dallas Heart Study.

The authors quantified the proteomic signature of plasma HDL in 36 individuals with persistent extreme cholesterol efflux capacity (CEC), defined as >=90th% (high) or <=10th% (low) over 15 years, using methods for analyzing proteins associated with apoA-I and quantifying plasma apoA-I containing lipoprotein subspecies. The main finding was heterogeneity in CEC-protein correlations between high and low CEC groups.

The results are interesting, and the manuscript is well-written and easy to follow. However, I have some questions that I would like the authors to comment on.

1. In association studies considering only the extremes of a continuous distribution, the results may be susceptible to bias due, among other things, to where the cut-off point is placed (which quartiles are compared). Considering only the extremes favors finding differences while losing sight of what happens in the middle zone of the distribution. The authors have yet to mention the possible limitations of their study.

2. There are studies in which, using subfraction determination techniques such as electrophoresis or nuclear magnetic resonance, they describe an increase in cardiovascular risk due to the decrease in a specific subfraction of HDL. Is there any correspondence between the AI-Lp subspecies and those determined by other techniques and associated with a lower risk of CVD?

Minor comments

1. The units in Table 2 must be in lowercase (mg/dL, g/dL).

2. The reported results in the abstract did not correspond to those in Figure 1. Please revise.

3. Abstract and page 5, lines 168 and 169. The r ranges include Alpha-2-macroglobulin (A2M). Please modify the ranges (r=0.58 to 0.65 and r=-0.11 to -0.25).

4. The first author of reference #10 is “AIM-HIGH Investigators,” not “Investigators, A.-H.” Please correct it.

5. Page 7, line 189. “CO3: Compement C3,…”. Please, type Complement C3.

6. The first author of reference #44 is “TG and HDL Working Group of the Exome Sequencing Project, National Heart, Lung, and Blood Institute,” not “Tg; Hdl Working Group of the Exome Sequencing Project, N.H.L.” Please, correct it. 

Reviewer 2 Report

The authors investigate the relationship between protein determinants of HDL in subjects with persistent high and low cholesterol efflux capacity. It has been well established that CEC as a measure of HDL function is superior to circulating HDL levels in CVD risk prediction, however much is still unclear why CEC is impaired or why contradictory high CEC is found in some patient cohorts. The questions addressed by this manuscript are therefore topical and relevant. Using various methods to determine apoA-I associated proteins this works indicates that protein levels are overall similar between high and low CEC groups but that the relationship between apoA-I associated proteins is different within each group and between apoA-I species with and without certain proteins. Although interesting, the use of different methods to determine apoA-I associated proteins and CEC are not clearly explained and referred to in the text of the manuscript. It is unclear how the methods compare and whether the results are consistent between the methods. Overall this work indicates that the mechanistic components of HDL determining cholesterol efflux are complex and need further investigation to how the presence of specific proteins on HDL affect CEC.

Comments:

The use of different methods to determine apoA-I associated proteins and the use of different methods determining CEC needs to be explained and consistently described throughout the manuscript, including the titles of sections and legends of figures and tables. In addition, findings using the different methods need to be compared and discussed. For example:

1.      Title 2.3 states that apoA-I subspecies display heterogeneous correlations with plasma CEC, however this sections mainly discusses fractionated (CEC-AUC).

2.      Section 2.3  states that correlation with plasma CEC were not observed. The plasma CEC data is not shown anywhere not is it discussed why correlations are found using CEC-AUC are not observed using plasma CEC.

3.      Supplement table 1 shows correlations between apoA-I associated proteins and plasma CEC. The title indicates that also apoA-I species with and without certain proteins are investigated. However, the text in the main manuscript only describes correlation between low and high CEC groups and not the data concerning the with and without species.

4.      Are similar relationships with CEC observed when apoA-I subspecies data with and without certain proteins are determined by mass spec or by ELISA?

5.      Supplementary Table 2 is referred to under the ELISA section. However, this method only investigates 7 associated HDL proteins. Where other proteins depicted in the table determined by Mass Spectrometry? If so, how does this data compare to the data in Supplementary Table 1.

6.      It is not clear from the legends to Supplement figures S2, S3 which methods for determining apoA-I subspecies are used.

7.      Why was the relationships of apoA-I subspecies determined by Mass Spectrometry only compared to plasma CEC and not fractionated plasma CEC (CEC-AUC)?

8.      The current data in the manuscript does not allow comparison between the relationship between apoA-I subspecies and CEC determined by the ELISA v Mass Spectrometry method.

Previous work (El-Ghazali et al ATVB 2021 41(10)2588-2597) used the same subjects and indicated that fraction-specific efflux was related to fractionated phospholipid content but no apolipoprotein A-I, cholesterol or total protein. Therefore a large part of the data described in this manuscript (subject characteristic, plasma and AUC-CEC measurements, no relationship with apoA-I) has already been shown. This should be clearly indicated.

Fractionation of HDL by FPLC has the advantage that findings can be related to different HDL-size classes as was done in the El-ghazali et al manuscript. Are there specific correlation observed for larger v smaller apoA-I subspecies in the current manuscript?

ApoA-I associated proteins were initially investigated using a his-tagged apoA-I “dipping” experiment. Several proteins were found to correlate with CEC in either low or high CEC groups but were not investigated further such as vitronectin, apoAIV or apoCII. These should be further investigated using ELISA to confirm consistency of the data.

The methods used examine apoA-I associated proteins only. It is known that apoA-I levels are decreased in certain patient groups and that HDL species not containing apoA-I exist and that levels of these can differ between cohorts. This should be mentioned in the discussion.

.

Round 2

Reviewer 2 Report

Thank you for the clarifications and amendments.